# Characterization of the Genetic Variability within *Ziziphus nummularia* Genotypes by Phenotypic Traits and SSR Markers with Special Reference to Geographic Distribution

**DOI:** 10.3390/genes14010155

**Published:** 2023-01-06

**Authors:** Nisar Uddin, Noor Muhammad, Sameh Samir Ali, Riaz Ullah, Ahmed Bari, Hidayat Hussain, Daochen Zhu

**Affiliations:** 1Department of Botany, Hazara University Mansehra, Pakhtunkhwa 21120, Pakistan; 2Research Center of Chinese Jujube, Hebei Agricultural University, Baoding 071001, China; 3Biofuels Institute, School of the Environment and Safety Engineering, Jiangsu University, Zhenjiang 212013, China; 4Botany Department, Faculty of Science, Tanta University, Tanta 31527, Egypt; 5Department of Pharmacognosy, College of Pharmacy, King Saud University, Riyadh 11451, Saudi Arabia; 6Department of Pharmaceutical Chemistry, College of Pharmacy, King Saud University, Riyadh 11451, Saudi Arabia; 7Department of Bioorganic Chemistry, Leibniz Institute of Plant Biochemistry, Weinberg 3, D-06120 Halle (Salle), Germany

**Keywords:** genetic resources, climatic change, genetic diversity, conservation

## Abstract

Understanding the impacts and constraints of climate change on *Ziziphus nummularia′s* geographical distribution is crucial for its future sustainability. In this study, we analyze information obtained from the field investigation, the distribution and response of climatic changes of *Ziziphus nummularia* by the use of ArcGIS analysis. The genetic diversity of 180 genotypes from three populations was studied by morphological attributes and simple sequence repeat (SSR). The results showed that the significant bioclimatic variable limiting the distribution of *Z. nummularia* was the mean temperature (bio 10_18.tif and bio19). Under the current climatic change, the suitable growth region of *Z. nummularia* is Swat (35.22° N, 72.42° E), while the future distribution would be Buner (34.39° N, 72.61° E), respectively. A total of 11 phenotypic traits were noted and had significant phenotypic variation among the traits. A total of 120 alleles were amplified. The alleles per locus ranged from 2 to 6, averaging 4.42, whereas PIC ranged from 0.33 to 0.79. Within a mean value of 0.67 per locus, expected heterozygosity was 0.57, observed heterozygosity was 0.661, and average gene diversity was 0.49. Flow estimates (6.41) indicated frequent gene flow within genotypes. The clustering, STRUCTURE, and PCoA analysis indicated Swat and Buner migration routes and evolution as well. The results indicated the prevalence of genetic variability and relationships among *Z. nummularia* across geographical boundaries had retained unique alleles. This may facilitate the development of agronomically desirable cultivars. However, climate change has impacted species distributions, requiring strategies to conserve genetic resources in different areas.

## 1. Introduction

Modern evolutionary processes, including gene flow, genetic drift, and natural selection, as well as a species’ evolutionary history, have an impact on the genetic diversity of populations and how they differentiate [1]. Due to the weaker genetic drift and less gene flow in small and isolated populations, genetic variation is frequently reduced, and genetic divergence between populations is enhanced [2]. Genetic diversity may be severely harmed by bottlenecks and founder effects [2]. Here, we focus mostly on the characteristics of the life history, historical environments, and climatic events that have created and will continue to shape the distribution of genetic variation within and among *Ziziphus nummularia* in Malakand division, Pakistan.

Data on population genetics based on molecular markers has been highly helpful in understanding historical occurrences in the past, and they will help assess how well-equipped species are to adapt to the current environment affected by climate change. However, a mixture of these events may also happen. Extant species that have survived through glacial refugia are likely to have genetic markers of within-refugium genetic drift and selection or, conversely, harbor historical genetic diversity not seen throughout a taxon’s range. For example, more and more suitable markers are available for measuring the recent genetic divergence within and among species to detect patterns for measuring population-level variation, such as microsatellites, restriction site-associated DNA markers (RADseq), and whole-genome sequences [3].

Previous analyses of the genetic diversity of *Z. nummularia* have primarily been based on the morphological tracks, but these indexes are more susceptible to different environmental conditions. With the great improvement of molecular markers technology, markers such as SRAP, RAPD, ISSR, AFLP, EST-SSR, and SSR are widely distributed in genotypes for the estimation of genetic diversity, relationship, genetic population, and genetic map construction, respectively, in the *Z. nummularia* genotypes. Genotypes of *Z. nummularia* provide strength for exploring and protecting the genetic and phenotypic diversity of breeding approaches. Its genetic diversity defines its potential to consistently produce new, high-quality varieties, which is crucial for the sustainability and advancement of breeding [4]. The basis for the collection, maintenance, study, development, use, and cultivar improvement of germplasm resources, as well as the heart of biodiversity [5], is genetic diversity. Due to their high polymorphism, codominant inheritance, excellent repeatability, and simplicity of application, SSR markers are very helpful in this research and have been frequently used to examine the genetic diversity of several species because of these advantages, including the pear, Taxus [6], jute, olive [7], and Jinsha pomelo [8]. Furthermore, the information on the natural diversity among these groups of genotypes is restricted and is less characterized at the molecular level. *Ziziphus* is a genus of around 200 species of spiny shrubs and small trees in the buckthorn family, Rhamnaceae. Species of the *Ziziphus* genus are distributed throughout the world in the warm-temperate and subtropical regions [9]. Plants of the *Ziziphus* genus are also highly adaptable, with outstanding drought, salinity, and high-temperature stress tolerance capacities [10]. The fruits of plants in *Ziziphus* are highly nutritious with valuable medicinal properties. Fruits are also a source of carbohydrates, minerals (iron and potassium), phenolic compounds, and vitamins [11]. Because of all these unique characteristics, the fruits of *Ziziphus* are considered superfruits.

*Z. nummularia* (Burm.f.) Wight & Arn., a perennial herb commonly called small ber and wild ber, which is native to southeastern Pakistan and India [12], is an important and well-known species of *Ziziphus* known for its nutritional, medicinal value [12], whereas, in traditional medicine system, different parts of *Z. nummularia* such as fruits, leaves, root, and tubers are the used for the treatment of various ailments, i.e., ulcers, pulmonary, fevers, wound healing, liver troubles, and asthma. It has cooling roots used to treat biliousness and headache [13]. Different online works of literary studies of *Z. nummularia* rewarded more consideration on genetic diversity, such as morphological and molecular diversity, as well as their identification, e.g., resistance to different diseases, stress, and high yield production. Limited studies were recorded on genetic diversity [13,14].

The development of human civilization and global climatic change has had a significant impact on the ecological environment. The spatiotemporal distribution of species is somewhat influenced by climate, and changes in species’ geographical distribution also reflect climate changes [15]. The geographical distribution, phenology, and other ecological indicators and developments of species are significantly impacted by climate change, which will accelerate the emergence or extinction of species [16]. It is critical to comprehend how wild species react to climate change to protect and manage wild species over time [17]. As a result, examining the relationship between climatic variables and species (including *Z. nummularia)* is essential to understanding how potential changes in a species’ geographic distribution and the effects of climate change on species survival are affected.

In order to help lead conservation programs, molecular studies on Pakistani *Z. nummularia* should be carried out to identify the species’ most likely origin and to locate centers of diversity [18]. The development of seed management strategies is also critical for increasing the chances of ex-situ conservation [18]. It may also be beneficial to examine or investigate the genetic variation in *Z. nummularia* genotypes found in Pakistan, as well as the identification of genotypes with significant differences in drought resistance, tree growth, and fruit yield [18,19]. Furthermore, *Z. nummularia* has enormous potential as an income and nutritional crop, particularly in Pakistan’s arid regions, and the estimation and screening of superior genotypes using a more prominent scientific method like SSR-based markers should be considered as a major research topic across the country.

Our understanding of the potential of wild plant species (*Z. nummularia*) and their unbiased diversity remains limited and requires expansion [18,19]. Moreover, wild plants like *Z. nummularia* have received little attention to date [20], so the primary goal of this study was to measure intraspecies genetic variation within a population of *Z. nummularia* adapted to different geographical regions. Various tools like morphological and SDS-PAGE seed storage protein have already been used to identify the diversity among different genotypes of *Z. nummularia* collected from Pakistan [21]. However, due to environmental fluctuations, morphological characterization was found unstable in the majority of cases [18,19]. Germplasm evaluation and genetic variation through DNA-based (SSRs) molecular markers is a very prominent and liable approach [22]. Simple sequence repeats (SSRs) have great potential in plant breeding and genetics; owing to them their codominant inheritance, abundance, genomic coverage, and robustness SSRs have become the marker of choice in studies of genetic diversity, identification of variety/species, molecular mapping, marker-assisted selection, and QTL analysis and negligible information is available of SSR markers of Pakistani *Ziziphus* species (*Z. nummularia*). Therefore, the goal of our current study was to reveal both the morphological and genetic variation and also the relationship of the *Z. nummularia* population in Malakand Division, Pakistan. Several studies have demonstrated that DNA-based molecular markers are valuable tools to unravel the genetic diversity of natural populations [22,23].

For that purpose, of these research limitations, many questions remain, such as the following: what are the levels of genetic diversity and genetic structure by the different ways of the existing *Z. nummularia* germplasms? How can the existing *Z. nummularia* germplasms resources be protected in the moving regions? With that perspective in mind, the combination of morphological and SSR markers was employed on 180 *Z. nummularia* genotypes belonging to three populations. Furthermore, this study used species distribution modeling to predict the present and future habitat suitability for the species, using distribution data and several environmental variables, to estimate its potential extinction risk due to climate change and to develop an effective management and conservation scheme. This study is expected to provide a more reliable basis for formulating protection and utilization measures for *Z. nummularia* germplasms.

## 2. Materials and Methods

### 2.1. Phenotypic Evaluation

Two consecutive samplings were arranged (2018–2019) within three districts [i.e., District Swat (35.2227° N, 72.4258° E), Dir Lower (35.1977° N, 71.8749° E,) and Buner (34.3943° N, 72.6151° E) of the Malakand division Khyber Pakhtunkhwa (KP) province (Figure 1).

Eleven phenotypic traits were recorded for 180 *Z. nummularia* genotypes collected from various locations of the Malakand division, viz., plant height (PH), branching (BR), leaf length (LL), leaf width (LW), leaf types (LT), petiole length (PL), internode length (InL), stem diameter (StD), fruit width (FtW), fruit diameter (FtD), and fruit length (FtL).

### 2.2. Genomic DNA Extraction

Total genomic DNA was extracted from the young leaves of selected genotypes using a modified CTAB (Cetyl trimethyl ammonium bromide) method [21]. Fresh leaves of *Ziziphus* species were ground to powder and placed into an Eppendorf tube where 800 μL of extraction buffer (3% CTAB, 5 M NaCl, 0.5 M EDTA, 1 M Tris–HCl pH 8.0) with 2 μL β-mercaptoethanol were added and incubated at 65 °C for 30 min. Incubation was followed by adding equal volumes of chloroform: isoamyl alcohol (24:1) before centrifugation at 12,000 rpm for 10 min. Genomic DNA from the aqueous phase was precipitated by using ice-cold 100% ethanol at −20 °C for 30 min and centrifugation at 12,000 rpm for 10 min. The supernatant was discarded, and the pellets were washed with 70% ethanol twice and centrifuged at 12,000 rpm for 5 min. The concentration and quality of the extracted genomic DNA were checked using spectrophotometers (Thermo Fisher Scientific, Waltham, MA, USA) and by loading 5 μL of DNA on a 1.0% agarose gel.

### 2.3. PCR Reaction for SSR Markers

A total of 27 pairs of SSR primers were applied to 180 genotypes of *Z. nummularia* (Table 1). These SSRs were selected from those of Xiao et al. [24]. The annealing temperature was set at 50–60 °C, and the optimal length of markers was 20bp, preferably with 50% GC content. Polymerase chain reaction (PCR) was performed in a volume of 12.5 μL containing 0.5 μL (~50 ng) genomic DNA, 6.3 μL of 2 × Taq Master Mix (CWBIO, Beijing), and 0.5 μL of 10 μmol/L each of forward and reverse primers. The reaction was set for 94 °C of denaturation for 3 min; 27 cycles of 94 °C for 30 s; 50–60 °C for 30 s; and 72 °C for 30 s; and a final extension at 72 °C for 10 min.

The unit consists of two clamp plates and allows the simultaneous running of two gels. The two gels are cast between glass plates with dimensions of 20 cm wide and 10.5 cm high for the front plate, and 20 cm wide and 11.3 cm high for the rear plate. The rear plate also has built-in spacers measuring 1.0 mm in thickness. Upon assembly, the final cast gel dimensions are 10.5 cm × 18.5 cm × 0.1 cm (H × W × T). Then, the gels were subjected to silver staining as described by Zhang et al. [25] for PCR products to be visualized using 8% denaturing polyacrylamide gel electrophoresis (PAGE) in 1×TBE buffer and run at 200 V for 50 min, followed by silver staining for 10 min (Trans Gen Biotech, China.

### 2.4. Data Analysis

The data analysis was performed by using different software and layers for the modeling and genetic diversity. The world-climatic dataset was used for the data, obtaining the bioclimatic layers of the selected study regions, while the composition of the layers was annual precipitation, mean diurnal range, annual mean temperatures, and pre-seasonality. For this, ArcGIS 10 software was used for thematic map-making and modeling.

The morphological data for the selected genotypes traits were further analyzed for various statistical parameters. They were analyzed for mean, maximum, and minimum by using the statistics software Excel 2016 and SPSS.

The SSR binary data obtained from the PCR amplification of the *Z. nummularia* 180 genotypes with 27 SSRs were assembled on a Microsoft Excel sheet and analyzed using POPGENE-32 software, including AF (allele frequency), PB (polymorphic bands), P (% present), A (% absent), He (expected heterozygosity under Hardy-Weinberg equilibrium), Ho (observed heterozygosity), GN (genotypes number), PIC (polymorphic information content), * na = the observed number of alleles, * ne = Effective number of alleles, * I = Shannon’s Information index, and FI = fixation index. In contrast, polymorphism information content (PIC) was computed with the following formula by Botstein et al. [26].
$$PIC=1-\overset{n}{\underset{j=1}{\sum }}P_{ij}{}^{2}$$
where *Pij* is the frequency of the *j^th^* allele for the *i^th^* locus, and summation extends over *n* alleles.

The population structural analysis is based on 27 SSRs analyzed with Structure Modal Software version 2.3.4 [27]. The optimal numbers of groups were determined by running a mixture and a frequency model related to the range of groups for the K value (the putative number of populations). Each execution consisted of a recording period of 10,000 steps followed by 100,000 simulations of MCMC (Monte Carlo Markov Chain). The choice of the most probable K value was made by calculating the estimated probability of data recording [LnP (D)] and ad hoc Δ K statistic based on the rate of change in LnP (D) between the successive K values [21]). The neighbor-joining assigned the 180 genotypes to the corresponding groups. Similarly, principal coordinate analysis (PCoA) is a visualization technique commonly used in multivariate statistics. In the neighbor-joining tree, a dendrogram is based on Nei’s genetic distance matrix with MAGA 5. Genetic variation within and between populations was identified by group analysis of allele frequency estimates using analysis of molecular variance (AMOVA). Pairwise estimates of correlations between inter-individual alleles (FST), inter-group fixed index (FIS), and individual internal fixation index (FIT) were calculated using the AMOVA method in GenAlEx 6.1 [28].

## 3. Results

### 3.1. Bioclimatic Layers

The current bioclimatic data were obtained from the world-climatic database for the application of Arc ESRI information. The differently formatted data were used, i.e., annual mean temperature, annual precipitation, mean diurnal range, and per-seasonality, to mention the area of the Malakand diversion KP, Pakistan. GPS data were recorded at the point where the collection was noted to be of different health statuses. Mature plants *Z. nummularia* genotypes were distributed into three different geographical regions. 

When a climatic variable was used alone, the three variables with the greatest value gain by the selection of three bioclimatic layers, which are superly imposed on the map and check the weather of selected regions, bio19 (Mean temperature and total precipitation) bio10_18.tif (Mean Temperature of Warmest) and bio11_18.tif (Mean Temperature of Coldest), respectively. This shows that the above three climatic variables were the main influencing factors for the prediction of the suitable distribution regions of *Z. nummularia* genotypes. The suitable distribution region of *Z. nummularia* genotypes in the Malakand division is in Swat and Buner. All the prediction results were predicated according to the distribution data of *Z. nummularia* in the Malakand division (Figure 2a,b).

The current results noted that the annual mean temperature from low to high and annul precipitation were designated by different color layers showing the distribution binderies of *Z. nummularia* genotypes and their biological origin (Figure 1B and Figure 2A). The mean diurnal range, precipitation seasonality from low to high, and overall temperature range of different color layers were designated to estimate the distribution of *Z. nummularia* into the selected geographical area of the Malakand division (Appendix A), respectively.

### 3.2. Phenotypic Diversity and Correlation among the Traits

Descriptive statistics of the morphological parameters are summarized in Table 2 and Figure 3. The %CV was calculated for plant height (PH), branching (BR), leaf length (LL), leaf width (LW), leaf types (LT), petiole length (PL), internode length (InL), stem diameter (StD), fruit width (FtW), fruit diameter (FtD), and fruit length (FtL), in genotypes was 90.55%, for ZNDR (Dir) was 54.86, and for ZNBU (Buner) was 11. Significant variation was found for the BR (Branching) among the genotypes of (3) regions. The highest value was observed for genotypes collected from ZNST (Swat) (69.46%), followed by ZNDR (Dir) (41.84 %), while the lowest value was recorded for the genotypes collected from ZNBU (8.766%). For LL in genotypes collected from (Swat) was (57.47%), ZNDR (Dir) (50.80%), and ZNBU (9.97%). Furthermore, the highest variation was observed for LW, LT, PL, InL, StD, and FtW. The number of FtL varied for genotypes collected from ZNST (56.10%), ZNDR (45.31%), and ZNBU (7.47%).

Furthermore, the Pearson correlation coefficient revealed a significant positive as well as a negative association (*p* = 0.05 and 0.01) among the studied traits of *Z. nummularia* (Table 3). Several traits revealed strong interrelationships within phenotype categories, particularly leaf traits with yield-contributing traits and a few traits correlating with other categories, such as inherently linked growth and phenology-related traits (Table 2 and Table 3).

### 3.3. Genetic Diversity and Population Structure

In the current research study, a total of 27 SSR pair primers were successfully amplified for 180 *Z. nummularia* genotypes. The size of the PCR bands ranged from 137 bp up to 240 bp (ZSSR−181 repeat motif as CTT-17 and ZSSR-460 as AG-11), respectively, in *Z. nummularia* genotypes (the list of these SSRs is expected to be band size and included in the experimentally obtained markers (Table 1)).

A total of 120 alleles were reported within 180 genotypes of *Z. nummularia* collected from three different regions of KP, Pakistan (27 SSR pair primers), and the mean allele was 4.43 per locus. The frequency of each allele at each SSR ranged from 0.510 to 1.190 in locus numbers ZSSR-513 and ZSSR-181, respectively. The lowest number of alleles were noted at the 11 loci ranging from 2 to 3 in (ZSSR-21, ZSSR-93, ZSSR-239, ZSSR-247, ZSSR-261, ZSSR-262, ZSSR-414, ZSSR-416, ZSSR-485, and ZSSR-490, respectively. For most loci, the difference in allelic size was observed to be almost completely repeated by the unit, indicating that the change in these sites is due to the difference in the number of repeating units. The observed genetic diversity, PIC (polymorphic information content), ranged from 0.33 at the locus of ZSSR-490 to a maximum of 0.79 at the locus of ZSSR-97. There were five different loci (ZSSR-95, ZSSR-152, ZSSR-175, ZSSR-188, and ZSSR-513) that were highly polymorphic, with PIC values higher than the other loci and where no locus was recorded with PIC less than 0.25, respectively. * ne = Effective number of alleles, * I = Shannon’s Information index, and *BP = Polymorphic bands were presented in Table 4.

The observed heterozygosity (Ho) in *Z. nummularia* ranged from 0.070 to 0.968 in the loci ZSSR-414 and ZSSR-152 with an average of 0.66, and the expected heterozygosity varies from 0.140 to 0.760 in the loci ZSSR-414 and ZSSR-95, respectively, with an average value of 0.575. The important information on the *Z. nummularia* genotypes, Viz, the effective number of alleles (* ne), and the Shannon information index (* I) are summarized in (Table 4). The observed number of alleles (* na) varied from 2 to 6 in the locus as ZSSR-490 and ZSSR-152 in *Z. nummularia* genotypes. The ** Nei’s ranged between 0.140 for locus ZSSR-414 and 0.758 for locus ZSSR-95. FIS varied from −0.110 to 0.816, with an average of −0.210. The FIS values obtained for most of the populations were not significant and negative (*p* > 0.05), which suggests that there is no loss of heterozygosity. FIT, FST, and Nm* (Table 4). Most of the *Z. nummularia* genotypes in population three showed abundant genetic variation, and the average gene diversity was 0.4944. The estimate of gene flow (Nm*) based on FST in all populations has an average value of 6.415.

The population structure analysis was carried out for the 180 *Z. nummularia* genotypes collected from three geographical regions of Malakand Division, KP, Pakistan, which were estimated by the user of the Bayesian clustering system probabilistically assigned individuals to respective populations. Three types of clustering were identified, which showed the relationships among them of different geographical origins. K values were estimated as the mean of the final estimations of L″(K) found the middle value of more than 20 runs isolated by the standard deviation of L(K), Δ K = m(| L″(K) |)/s[L(K)] (Pritchard et al. 2000). For the given ranges, i.e., 1 to 20 and the highest values were recorded in K = 3 and K = 5. Consequently, the analysis of the STRUCTURE software was performed for the K = 3, which was largely grouped by the genotypes.

According to Pritchard et al. (2010) and Bayes’ rule estimation of K, values give a comparative estimation of LnP(D), and the smallest value is regarded as almost correct. The estimation of ΔK values that the peak reached K = 5. According to the formula, our result shows clear peaks of ΔK at K = 3 of *Z. nummularia* genotypes (Figure 4). In the absence of clear-cut origins of the accessions, a non-stratified strategy was adopted for the genetic structure analysis.

Our results showed a clear peak for ΔK at K = 3, where all the accessions were roughly divided into three major groups, with some admixture among groups (Figure 4). The ΔK value was recorded at K = 3, which suggested that the 180 *Z. nummularia* genotypes were approximately divided into three CPs (CP = 1, CP = 2, and CP = 3), accordingly (Figure 4 and Figure 5). The genetic relationship among the CPs provided various confirmations of gene flow between CPs. CP = 2 consisted of the highest number of genotypes (63), followed by CP = 3, having 44 genotypes recorded, and CP = 1 contains 27 genotypes of *Z. nummularia*. CP = 3 was comprised of almost all of the ‘ZNBU included in CP = 2, while the collected genotypes from ZNDR of the *Z. nummularia* were assigned to CP = 2 and CP = 3. Notably, they were highly adaptable to moderate climates and were included in CP = 2 and CP = 3, suggesting that they may have a unique ancestry type. Statistical analysis indicated that the percentage of genotypes with a membership coefficient ≥ 67% was 45.00%. A total of 60.00% of genotypes exhibited a membership coefficient ≥ of 73%, and only 1.99% of the genotypes exhibited a membership coefficient of 3.2% or less. Based on standard permutation tests of the full data set, the groups defined by *structure* suggest moderate genetic differentiation, as indicated by the global FST mean value of 0.038 (*p* < 0.01). Pairwise FST comparisons among the different geographical regions showed that FST values varied from 0.0013 to 0.1913, respectively (Table 5).

### 3.4. Neighbor-Joining (NJ) and Principal Coordinate Analysis (PCoA)

The neighbor-joining system was used for the composite phylogenetic tree for the 180 *Z. nummularia* genotypes collected from three geographical regions. The results of the STRUCTURE test largely agreed with the results of the ZNST, ZNDR, and ZNBU (Swat, Dir (L), and Buner) of KP, Pakistan. The analysis showed that, in general, the genotypes from the different regions clustered together represented three clusters, i.e., Cluster I, Cluster II, and Cluster III (Figure 6). Cluster I was the largest group, which comprised mainly the ZNBU, 16 genotypes from ZNST regions, and 27 genotypes from the ZND region. Cluster II contained a total of 40 genotypes, 9 from ZNBU, 03 were ZNDR, and 28 were from ZNST regions, and Cluster III comprised most genotypes from ZNST (Swat) and ZNBU (Buner). The 54 *Z. nummularia* genotypes were separated into two sub-clusters within the cluster (Figure 6). Sub-cluster 2 comprised only ZNST, while *Z. nummularia* genotypes sub-cluster 1 included ZNBU and ZNDR, respectively, and there is very strong support for clustering genotypes with related geographical origins.

The PCoA (Principal coordinate analysis) was divided into three clusters (Figure 7), which contained the assignments composite by the ZN clustering method and population structure (Figure 5 and Figure 7), respectively. The majority of *Z. nummularia* genotypes belong to Cluster I. ZNST genotypes were distributed in the left half of the PCoA plot, while the rest of the *Z. nummularia* genotypes from district ZNBU and ZNDR regions belonging to Clusters II and III are distributed on the right of the PCoA plot. The distribution of Cluster I was more widely scattered than Clusters II and III, indicating that the *Z. nummularia* genotypes of the Swat region had higher diversity than the genotypes of the Buner and Dir regions. The PCoA results corresponded to cluster analysis as the Swat genotypes were close to Dir, and the Buner genotypes were joined with the Buner region of *Z. nummularia* genotypes.

### 3.5. Analysis of Molecular Variance (AMOVA)

The genetic diversity was recorded within and among the population of different geographical regions and population structures, providing arranged data for the hierarchical AMOVA (analysis of molecular variance). The current total genetic variance among the population was 8%, while differences among individuals within populations contributed 12% to the total variance, and 72% of the total variance occurred within individuals.

The determination of FIS values as 0.102, FIT showed the highest level of genetic variation at 0.105 and FST values of 0.092, respectively, suggesting that the fixation index value of the *Z. nummularia* genotypes was highly significant (*p-Value* 0.001) (Table 5).

## 4. Discussion

### 4.1. Distribution and Climate Change

*Z. nummularia* and forecast an especially uncertain future for this rare species of *Ziziphus*, which is native to Malakand division, KP, Pakistan. It is looking likely that the *Z. nummularia* is going to harshly reduce ranges, as recommended by the up to 99% decreases of the suitable habitats across all the global climatic models during these periods. Due to the results, the plant species are more likely to face a high risk of extinction by local restrictions. Climatic change has already affected several plant species which were distributed all over the world. Mostly tropical and subtropical regions of the world have been identified as the most sensitive regions in Pakistan, where the projection percentages of species loss will reach 60% by 2080 [23,29].

As Hindukush regions range in the northern regions of Pakistan, many plant species, one of the *Z. nummularia*, are likely going to face a wave of mass extinction of the Z. *nummularia* genus over the coming periods. However, the high level of genetic diversity of the current *Z. nummularia* genus may eventually favor adaptation and perseverance but remain highly ambiguous. Species growth will depend on factors such as genetic makeup, fitness, healthy habitat, and fragmentation while dispersal approaches. The seed of *Z. nummularia*, which lacks obvious adaptations for long-distance and brochure dispersal, is most likely the primary mode of dissemination. It necessitates the use of crude methods, such as grazing mammals that consume the seeds along with the foliage, which cannot be ruled out.

### 4.2. Morphological Variation in Z. nummularia Genotypes

The total of 11 quantitative morphological traits are often known to be influenced by environmental factors, but the most important of these traits cannot be underestimated for analyzing the diversity of a *Ziziphus* species, mostly for *Z. nummularia* genotypes and crop species, as they are the primary constituents of overall diversity. Different types of systems, methods, and researchers have been utilized for morphological variation in combination with molecular markers to precisely estimate characteristic variability for drawing inferences in many crops and trees, which include *Ziziphus* species (*Z. nummularia*) [23,30].

In the current study, the information generated through multivariate analysis and genetic parameters has given an excellent view of genetic diversity for the 11 quantitative traits for the 180 *Z. nummularia* genotypes collected from different regions (Swat, Dir, and Buner) of KP, Pakistan. The level of variation was enough to discriminate against many genotypes of *Z. nummularia* from each other in our study. This agreed with many earlier studies involving morphological traits that distinguished different genotypes among the three regions of KP, Pakistan [22,31,32]. High LL (Leaf length) suggests for ZNST, ZNDR, and ZNBU that these traits may be under the influence of additive gene interactions, and the use of the simple selection methods (single replication, single plant selection) would be sufficient for further improvement of these traits [33]. Correlation coefficients are important in plant breeding because they measure the degree of association (genetic and non-genetic) between two or more traits. In the presence of a high correlation between different traits, selection for a particular trait will cause a change in its mean through the additive gene effects of selected individuals and simultaneously cause an indirect change in the mean of the other trait [12]. The significant positive correlations of FtW with LL, LW, and FtL represented that they showed that selection for any of these traits might favor improvement in fruit production.

### 4.3. Genetic Differentiation and Population Structure Analysis

An essential sign of a species’ capacity to withstand changes in its environment is its genetic diversity. Selection, genetic drift, and breeding techniques all have an impact. The genetic diversity of Z. nummularia genotypes may be studied to give a theoretical framework for preserving and employing Z. nummularia genotypes.

The results (He = 0.281) obtained by Zhang [34] using ISSR markers to analyze 72 samples in four major production locations were comparable. Based on SSR markers, the average He of 203 *T. sebifera* samples from eight populations was 0.486 in this study, which was greater than that of earlier research. Two possibilities could account for this discrepancy. First, different marker kinds were employed. SSR markers can reveal greater genetic variation than ISSR markers. Additionally, there were variations in the number of samples. The selection of samples for this investigation was fairly extensive. The primary *T. sebifera* distribution areas were also included in the distribution range. Six pairs of SSR markers were employed by DeWalt et al. [34] to analyze 129 samples from 12 *T. sebifera* populations in China, and the values were higher than those of this study (average He = 0.70), which was quite low for our current results.

For the genetic diversity using various parameters of the SSR loci, viz. the mean values of allele frequency, allele numbers, observed and expected heterozygosity, and PIC as well is important to reveal the genetic diversity of *Z. nummularia* genotypes [35]. These parameters were found to be high in the current study. An average of 4.4286 alleles per locus was higher than in the previous studies conducted on the different plants [35,36]. Similarly, the average PIC value recorded in our study was almost 0.671, which was greater than the average PIC obtained by Ponnaiah et al. [34], Ahmad et al. [36], and Cieslarová et al. [37]. However, in another study carried out by Jing et al. [30], higher PIC values were obtained by using SSR markers. Likewise, the mean heterozygosity value was higher than that detected in the studies done by other workers [38]. The average Shannon’s Information Index (I) value of 1.0339 was also almost similar to what was observed by different citations and coworkers of Cieslarová et al. [37], who reported an average I value of 1.22 with SSR markers and an average I value of 0.59 using retrotransposon markers in pea cultivars.

High estimates obtained for all diversity parameters indicated allelic richness in the analyzed genotypes, which can be utilized in breeding programs to get the desired plant types for breeding. While 120 total polymorphic alleles were recorded, the highest alleles were six per locus, while the lowest two alleles were noted with an average value of 4.4286, respectively. This information can be helpful for future genetic analysis specific to linkage groups. The analysis of molecular variance (AMOVA) results showed major and significant (80%; *p* = 0.001) contributions of within-individual differences to the total variation, whereas among-individual and among-population differences contributed 12 and 8%, respectively.

For the population, the structure performs a Bayesian genetic and admixture by modal STRUCTURE software and assigns all *Z. nummularia* genotypes of all three different geographical origins into three structural genetic clusters. In addition to assigning individuals to different clusters based on allele frequencies, it also detected that the extent of admixture of different genotypes collected from three regions which were shown in the three genetic clusters based on STRUCTURE, clusters CP2 broadly contained *Z. nummularia* genotypes collected for the Dir region, (ZNDR), respectively, the CP1 and CP3 were different types, which were a mixture of ZNBU and ZNST region genotypes. This clustering of pea accessions agreed with the study carried out by Jing et al. [30], who also showed robust clustering at K= 3, although the germplasm analyzed by them contained different species and subspecies. The mixing seems dominant over factors such as the self-pollination behavior of the crop and the wide geographical range/barriers. One of the striking observations of the STRUCTURE analysis was that the genotype of ZNDR represented no admixture with the ancestries, with the genotypes of ZNST and ZNBU beginning differently from those of other pure ancestries. This indicates that these were probably the regions from which a large amount of dispersal of genotypes has taken place in previous times.

The molecular phylogenetic and PCoA analysis were applied for the identification of *Z. nummularia* genotypes, and they were found in ideal molecular markers for the making of molecular phylogenetic relationships in *Z. nummularia* genotypes of the different geographical origins of KP, Pakistan. The N-J Phylogenetic tree distributed all *Z. nummularia* genotypes into three clusters, which was in correspondence with our previous studies and reported studies (38–39], which also reported the same number of groups from different plant genotypes for different geographical regions. Furthermore, the assignment of genotypes into groups appeared independent of geographic origins. However, Cluster I and Cluster III contain most of the genotypes collected from the ZNST and ZNBU regions of KP, while Cluster II includes those of the ZNDR region of *Z. nummularia* genotypes from distant geographical locations, which obstructed their specificity to geographic origins. Similar results were obtained by Ahmad et al. [33] in cluster analysis, where they found intermixing of *Z. nummularia* genotypes in different groups. When analyzed concerning gene pools detected by STRUCTURE, the groupings of the N-J tree appeared stronger with a few exceptions. In Cluster I, 17 genotypes from Cluster I from the Dir region, 30 genotypes from Cluster III from the Buner region, while in Cluster II, 3 genotypes from (ZNST 1, ZNST 9, and ZNST 43), respectively, and 9 from Cluster I (ZNBU 1, ZNBU 29, ZNBU 33, ZNBU 3, ZNBU 9, ZNBU 43, ZNBU 7, ZNBU 19 and ZNBU 32), respectively. Cluster III has a mixture of all three regions, 17 genotypes from Cluster I and 15 from Cluster II ZNDR. From this, it can be inferred that the clustering of *Z. nummularia* genotypes was based on genetic stocks regardless of geographic proximities. This could be conveniently ascribed to the movement of germplasm from one location to another in ancient times, the records for which are missing, and presently formal germplasm exchange agreements are in place between different regions as described by Kuleung et al. [34] and Ahmad et al. [33]. Thus, the present study further suggests that, despite its self-pollinating nature, there is sample intermixing of *Z. nummularia* genotypes collected from KP, Pakistan, which has resulted in maintaining diversity to a greater extent than can be surveyed and utilized by breeders to develop new, improved varieties in *Z. nummularia*. The N-J results were complemented with PCoA analysis.

### 4.4. Analysis of the Evolutionary Relationships of the Z. nummularia Population

The genetic diversity of plant populations is closely related to the climatic variability of their habitats [38]. The process of species migration or distribution and adaption to new habitats must be accompanied by a change in climatic conditions, which will promote the genetic diversity of species. The evolutionary relationship of the three populations of *Z. nummularia* in Dir (L) may be related to the change of habitats during the migration of the species. This analysis revealed that the genetic identity of *Z. nummularia* between populations/districts was negatively correlated with geographical distance, implying that *Z. nummularia* may have undergone a pattern of isolation by dispersal limitation [39].

### 4.5. Conservation Implication

The global application of molecular markers (molecular genetic diversity) technologies is attracting more and more common practices to the estimation and study of many aspects of plant genetic resources management [40]. The current study’s SSRs markers analysis was used to validate taxonomic origin and to determine the distribution of genetic diversity across the collection area of the wild *Ziziphus* species. With this investigation, *Z. nummularia* has been regarded as an endemic species. Furthermore, a significant range contraction (about 100%) is predicted for this species by the year 2050, even though no seed accessions from this species have been deposited in any global germplasm collections. In order to ensure species survival, a combination of ex-situ and in-situ conservation measures is required.

The field assessment of the collection revealed some interesting traits useful for the *Ziziphus* species’ genetic improvement [40]. Achieving genetic gain through conventional breeding takes a very long time for *Z. nummularia* because approximately four to five years is required to complete one cycle of field data collection and evaluation for the species. Therefore, appropriate planning to preserve individual palms that possess traits of interest should be in place to ensure long-term accessibility. The values of the current potential collection where determined genetic diversity is preserved in a reduced land area can be applied. The establishment of such a collection is an ideal option for oil palm as it increases the efficiency of conservation and allows for more effective access to genetic materials. The results presented above facilitate the identification of unique populations or rich allelic individual palms as well as populations that exhibit high genetic variation.

The present findings strongly suggest that there is wide genetic diversity present in the wild within genotypes of *Z. nummularia* collected from the area of KP, Pakistan. A conservation approach addressed to preserve single outstanding individuals is necessary, namely for the importance of heritage trees, which often become one of the few objects of great natural value in areas under the impact of land degradation and the falling into the decay of rural areas [41,42,43,44,45]. Knowledge of genetic variation within and among populations provides information essential in the formulation of appropriate management strategies for conservation [42,43,44,45,46,47]. The natural areas where the species of interest occur may provide relevant information to develop sampling strategies that will maximize the probability of collecting genetically distinct samples [45]. Based on the results of the present study, the probability of sampling identical or very similar genotypes of *Z. nummularia* increases when sites are sampled less than 20 km apart. Even though very distinct samples, albeit at a low frequency, can be obtained from the same site, a distance of 20 km between collection sites could be used as a general guideline in future collection missions to maximize the diversity captured for *Z. nummularia*. While the current result shows that the Pakistani *Z. nummularia* genotypes are exceptionally diverse and variable, this can be endorsed to the diver’s neighborhood genetic basics, specific reproducing weights, and the restricted trade of hereditary material [34,46]. The exceptional nature of the Pakistani *Ziziphus* species, uncovered by our results, backs the case for the execution of more extraordinary characterization and preservation procedures and potential rearing for *Ziziphus* species (*Z. nummularia*).

## 5. Conclusions

The effect and the incensement of global climate change in recent years, while the reason for climate change is heavy rainfall has caused floods (disasters) in various areas, and weather stations over the world measured the increasing concentration of atmospheric CO_2_ up to 400 pp, the highest in millions of years. Climate changes have exceedingly high effects on plants and wildlife, respectively. In the current research, we accompanied a complete study of genetic diversity, the population structure of *Z. nummularia* collected from Malakand division KP, Pakistan, to study diversity based on morphology and SSR markers. During the current study, a total of 180 genotypes of *Z. nummularia* were evaluated with a significant positive correlation. Descriptive analysis showed a high coefficient of variation between the study’s morphological traits of *Z. nummularia* genotypes. To date, this is the first time SSR fingerprints for 180 genotypes of *Z. nummularia* have provided comprehensive data on genetic diversity, population analyses, and their relationship with their geographical/climatic distribution were addressed. A total of 120 polymorphic alleles among the 180 genotypes were assessed by using 27 SSRs as reproducible markers. A visual understanding of plants’ genetic variation is important to conservation and management efforts. Remarkably, wise utilization of conserved plant germplasm in a gene bank collection needs knowledge of the genetic diversity that is present as well. Ensuring sustainable food security is a multi-faceted challenge involving much more than just increasing food production; it is also about protecting the useful diversity of wild landraces. The results shown here may provide valuable resources for future *Ziziphus* breeding programs. Furthermore, the application of the recently evolved high-throughput platforms and phenomics tools could enable plant breeders to identify and exploit the available plant genetic resources in environmentally sustainable ways. In brief, the current result will be highly beneficial in designing studies relating to *Ziziphus* species (*Z. nummularia*) genotypes’ phylogeny, as well as finding the biological origin and formulating future breeding programs.

## Figures and Tables

**Figure 1 genes-14-00155-f001:**
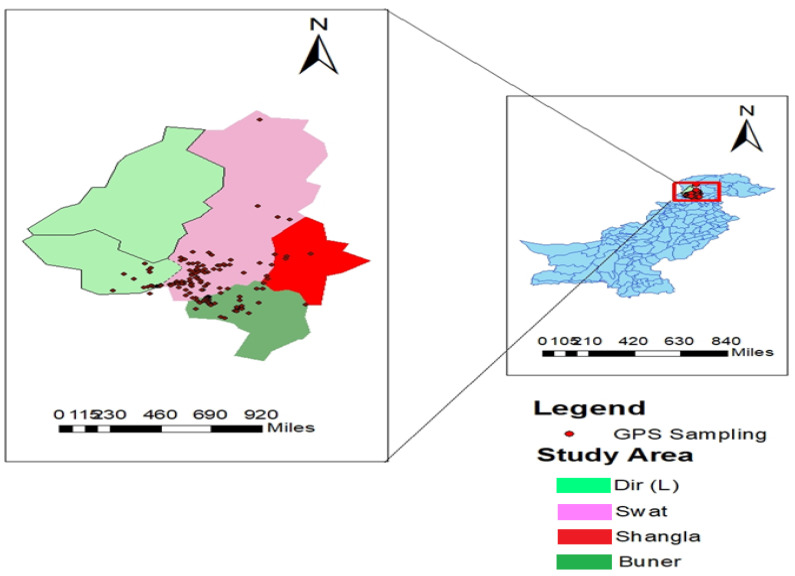
Map of the study area highlighting points of genotypes collection within Malakand division KP, Pakistan.

**Figure 2 genes-14-00155-f002:**
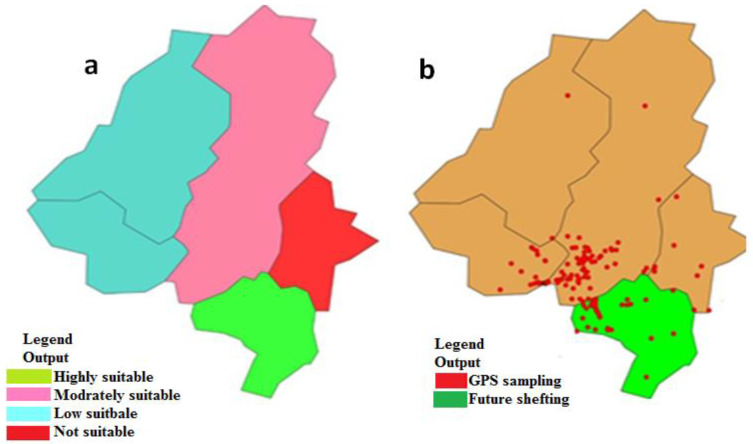
Showed that the ArcGIS mapping of a bioclimatic variable on the selected regions, (**a**,**b**) designated for the suitable distribution regions of *Ziziphus nummularia* (**A**), annual mean temperature and (**B**), annual precipitating.

**Figure 3 genes-14-00155-f003:**
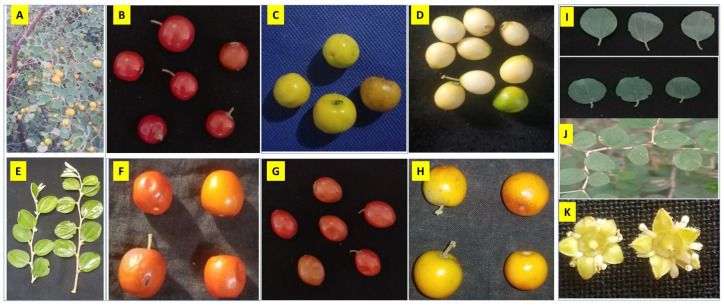
Photographs of collected *Z. nummularia* fruits and leaves from Malakand Division, KP, Pakistan (**A**–**H**) designated diversity in fruits shape/size and color and (**E,I,J**) designated leaf diversity and (**K**) for flowers of *Z. nummularia*.

**Figure 4 genes-14-00155-f004:**
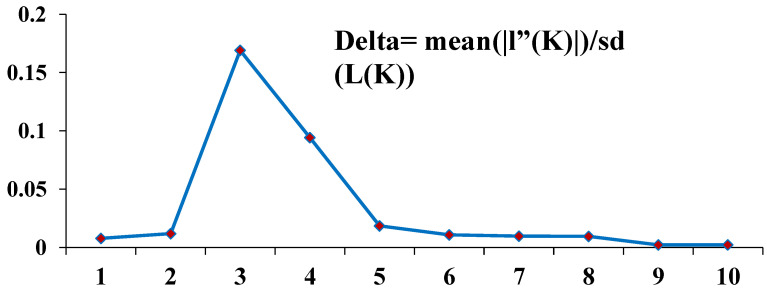
The Delta K values for the analysis STRUCTURE of *Z. nummularia* genotypes were calculated according to Evanno et al. (2000) and are plotted against the number of modeled gene pools (K).

**Figure 5 genes-14-00155-f005:**
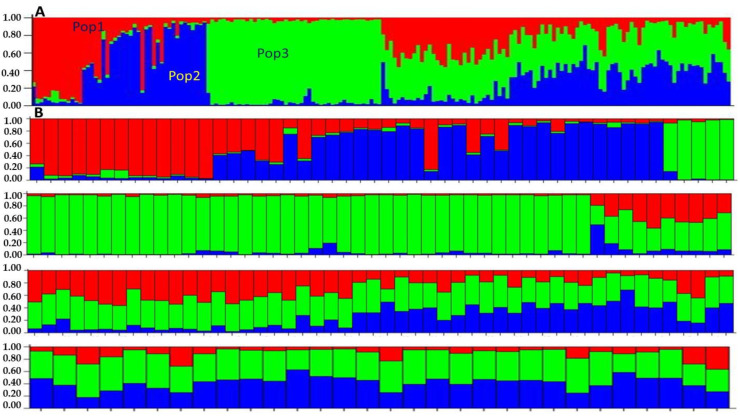
Population structure analysis: the population of 180 *Z. nummularia* genotypes with K = 3 clusters based on 27 SSRs primers. Note; the sub-figures represented relationship with other districts genotypes respectively.

**Figure 6 genes-14-00155-f006:**
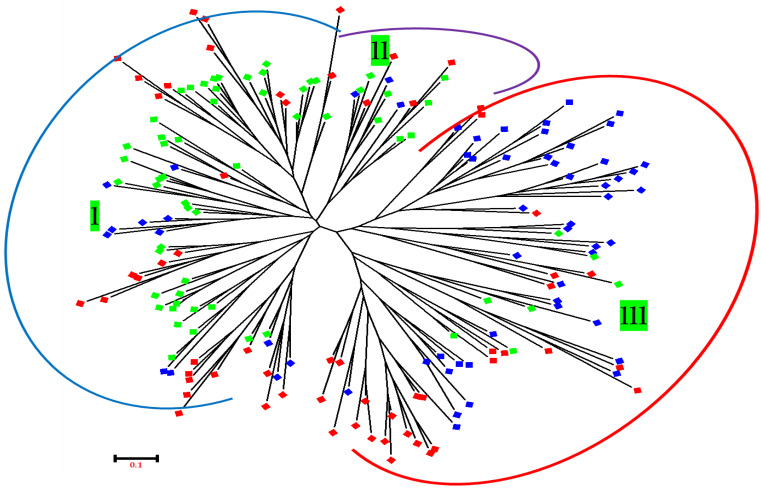
Neighbor-joining (N-J) phylogenetic represented genetic relationship among the *Z. nummularia* genotypes collected from different geographical regions of KP, Pakistan. Note: Red was designated for District Swat, green was designated for District Dir (L), and blue was designated for District Buner.

**Figure 7 genes-14-00155-f007:**
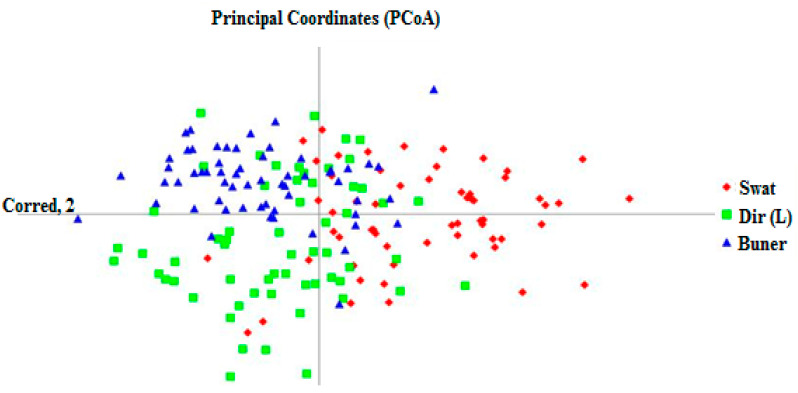
Principle coordinate analysis (PCoA) plots to show the clusters among 180 *Z. nummularia* genotypes.

**Table 1 genes-14-00155-t001:** Description of 27 SSR pair primes used for *Z. nummularia* collected from different geographical regions KP, Pakistan.

Locus	Repeat Motif	Forward	Reverse	BP	Temp °C
ZSSR-21	AT(21)	ACTCATTCCGTAAATTACACAGCC	TGAATTTCTAAATTTCACCAAAAACAA	228	57 °C
ZSSR-93	TTG(13)	GGAAGGACTTTGTCAGCATGGTAG	AACAGCATATTTGGATCCATTTCG	155	57 °C
ZSSR-95	TTG(11)	CGGTGAGAGACATTTTGTTGGATT	TTCCTTACTTTCCCACCTTGTTCA	152	57 °C
ZSSR-97	TTG(9)	GTCCAAAGGCCCAACTTCTTTAGT	AGGGGACTACTCCTCTGCTGAGAT	155	57 °C
ZSSR-110	TTA(15)	ACCTTTCGCTTAGTTCTTGCTGAA	GGACTTTTGTCCGGACCTTAAAA	225	57 °C
ZSSR-129	TGG(9)	TGCTAATGAAAGGAACTCTGGGTC	TGATGGGTATGAAGAAGCATCAGA	158	57 °C
ZSSR-152	TAT(18)	TCAGAATTTCTCACTTTGGCTAACAA	TGCACCGATCCTCTCCTCTC	160	56 °C
ZSSR-175	GTT(9)	CAAAGGAAAATCTCTATGGTTGTCG	CGCTACCATGTTAAAATTTGTCCC	155	57 °C
ZSSR-181	CTT(17)	TTTTGTCTCTCCCTCTTTTCTCCA	GGCCTTTTCATGAAGCTTTTGTTA	137	57 °C
ZSSR-183	CTT(14)	ATAGCAGCAATGGCTTTTTCTTTG	TTGAATTCCATGACATGAAGGTTG	156	57 °C
ZSSR-188	CTT(11)	ATCTCGTTAGTGCCTATCACCAGC	ACAGGAACATAAAAGAAGAAGAACGC	160	57 °C
ZSSR-192	CTG(9)	GCAGTTCTACATCATTCCCTCCTC	GAGATAGCATATCTGTTGGGTGGG	157	56 °C
ZSSR-207	CAT(8)	CCAAACACCAACCTTGTAATTGGT	TGTTCATGGAGACGATAAATCTGTTACT	160	55 °C
ZSSR-209	CAG(8)	AATGATCATGGGGAAACCAGTAGA	CTTCACTGCTCTGTTTGCTGTTGT	153	57 °C
ZSSR-239	ATG(12)	GCAAGTACCATACACAGGATACGTC	GCATAAAGTTTGTGGAAAACGTAATTT	158	58 °C
ZSSR-222	ATT(17)	GCAGCTGGATGAGAACCATAA	ACAATACAATACAAAGCCACATTAGTTC	146	58 °C
ZSSR-247	ATC(11)	CGCTACAAGTGTGCGATTATGAAA	GCCCAAAAGCTACAATACCTACCC	201	58 °C
ZSSR-261	ATA(10)	CTTGTCCAAAAGCTAACATACTTGC	AGGACAGTTCAGTAGGGTTTGTATTATT	238	56 °C
ZSSR-262	ATA(9)	CGTGGACCAAGTCTATACCAAAATG	TGGTTTTTCTTCTCCTAATCCATGTG	240	56 °C
ZSSR-414	AC(30)	TGCCACTAAAACGTATCCAATCAA	GGCATATCTGCTGAGGTGTATGTG	232	56 °C
ZSSR-416	AC(26)	AGGGCCAAACAAATTACAAGGATT	TTGGATGTTGAAGCTGTTTTACCTC	151	57 °C
ZSSR-460	AG(11)	GTTGTTCTTTTGGACCAATCCAGT	TGCATAGAACACCTTAGACAATGGAA	240	56 °C
ZSSR-465	CA(25)	GGTTGTGCTCATATGGGGATGTAT	TCTCCAAGATCCTTTTTGTTTTGC	220	57 °C
ZSSR-485	CT(28)	GTGCATGCATAAAAATCAAAACGA	GGGGTTTTATAGAAAGAGCGTGGT	186	57 °C
ZSSR-490	CT(23)	TTTGCTTCTTGGCTTCGACTAAAG	AGCCTACACAAAGGACTCTTTCCA	180	57 °C
ZSSR-513	GA(25)	TTCCTACCCCATCTGTACCTACTGT	TGGATGAATGATTGAAATGAAAAA	223	57 °C
ZSSR-774	AGAAA(5)	ATTGGGTTAGTCGAAAAATGGTCA	CCAATCTACAAGTGCTATGAGGCA	153	57 °C

**Table 2 genes-14-00155-t002:** Descriptive Statistics of *Z. nummularia* genotypes collected from different geographical regions of KP, Pakistan.

Swat							Dir (L)					Buner				
Traits	NO	MN	MX	Mean	Std. Devi-	% CV	MN	MX	Mean	Std. Devi-	% CV	MN	MX	Mean	Std. Devi-	% CV
PH	60	3.7	75.2	14.73	13.34	90.56	3	25	8.6	4.72	54.87	3.5	5.5	4.36	0.5	11.46
BR	60	2	38	9.17	6.37	69.47	2	16	5.73	2.4	41.84	13.5	20	16.38	1.44	8.77
LL	60	11.8	83	32.65	18.77	57.47	11.2	62.8	22.65	11.51	50.81	7.57	11.57	9.15	0.91	9.97
LW	60	8.8	55.4	24.34	13.58	55.78	5.8	40.6	14.97	6.85	45.76	5.17	7.83	6.46	0.74	11.47
LT	60	0.13	4.45	0.42	0.34	80.83	0.11	0.6	0.32	0.12	36.46	3.6	6.17	4.79	0.65	13.58
PL	60	2.8	20.6	7.38	4.39	59.48	2.2	34.8	7.25	5.54	76.45	2.5	3.97	3.46	0.37	10.76
InL	60	2.18	65.4	17.27	10.45	60.5	1.42	25.2	13.84	5.4	39.05	1.67	5.33	3.53	0.73	20.75
StD	60	1.1	21.6	4.12	3.25	78.95	1	14	3.19	2.07	64.9	9.87	13.03	11.27	0.68	6.06
FtW	60	65	148	104	18.83	18.14	58.02	230.6	1.11	27.12	24.37	22	32	28.26	2.57	9.08
FtD	60	4.8	334.8	25.26	41.9	165.87	4.2	28.8	9.71	4.32	44.45	3.67	15.53	4.73	2.04	43.18
FtL	60	5.2	48.6	20.66	11.59	56.11	6.4	35.2	12.45	5.64	45.32	206.33	303.67	2.7	20.22	7.48

Note: Plant height (PH), Branching (BR), Leaf length (LL), Leaf width (LW), Leaf types (LT), Petiole length (PL), Internode length (InL), Stem diameter (StD), Fruit width (FtW), Fruit diameter (FtD), Fruit length (FtL), NO (Number of genotypes), MN (Minimum), MX (Maximum) and CV (Coefficient of variance).

**Table 3 genes-14-00155-t003:** The correlation coefficient among *Z. nummularia* genotypes collected from different regions of KP, Pakistan.

Swat (ZNST)											
	PH	BR	LL	LW	LT	PL	InL	StD	FtW	FtD	FtL
**PH**	1.00										
**BR**	−0.04	1.000									
LL	0.386 **	0.395 **	1.000								
LW	0.492 **	0.258*	0.859**	1.000							
LT	−0.03	0.111	0.099	0.144	1.000						
PL	0.07	−0.066	−0.122	−0.144	0.166	1.000					
InL	0.10	−0.111	−0.335 **	−0.341 **	−0.144	0.255	1.000				
StD	0.681 **	−0.188	0.111	0.268 *	0.000	0.233	0.099	1.000			
FtW	−0.08	−0.077	**−0.332** **	−0.255	0.166	0.099	0.066	−0.022	1.000		
FtD	−0.04	0.044	−0.022	0.022	−0.033	−0.122	−0.211	−0.022	−0.166	1.000	
FtL	0.20	0.366 **	0.341**	0.368 **	0.111	−0.188	0.022	0.166	0.199	0.144	1.000
	**Dir (ZNDR)**										
	**PH**	**BR**	**LL**	**LW**	**LT**	**PL**	**InL**	**StD**	**FtW**	**FtD**	**FtL**
**PH**	1.000										
**BR**	0.144	1.000									
**LL**	0.044	0.301*	1.000								
**LW**	0.155	0.233	0.862 **	1.000							
**LT**	0.233	0.177	0.011	0.044	1.000						
**PL**	0.066	0.022	0.565 **	0.656 **	−0.088	1.000					
**InL**	0.111	0.155	0.022	0.000	−0.011	−0.244	1.000				
**StD**	0.529 **	0.364 **	0.313 *	0.330 *	0.177	0.200	−0.045	1.000			
**FtW**	0.000	0.077	0.278 *	0.340 **	0.055	0.166	0.133	−0.200	1.000		
**FtD**	0.099	0.255	0.511 **	0.559 **	−0.324*	0.547 **	−0.244	0.200	0.299 *	1.000	
**FtL**	−0.022	0.077	0.416 **	**0.469 ****	−0.322*	0.482 **	−0.199	0.022	0.438 **	0.785 **	1.000
	**Buner (ZNBU)**										
	**PH**	**BR**	**LL**	**LW**	**LT**	**PL**	**InL**	**StD**	**FtW**	**FtD**	**FtL**
**PH**	1										
**BR**	−0.012	1.000									
**LL**	−0.079	**0.475 ****	1.000								
**LW**	0.237	0.303 *	0.298 *	1.000							
**LT**	0.148	**0.508 ****	**0.397 ****	0.329 *	1.000						
**PL**	−0.037	−0.076	−0.065	−0.263 *	−0.135	1.000					
**InL**	−0.205	0.117	0.19	−0.029	0.032	0.089	1.000				
**StD**	0.149	−0.021	−0.016	0.024	−0.260 *	0.061	−0.174	1.000			
**FtW**	−0.163	−0.018	0.113	0.072	0.207	−0.093	−0.104	−0.300*	1.000		
**FtD**	**0.364****	−0.194	−0.129	0.132	0.005	0.149	−0.329 *	0.051	0.233	1.000	
**FtL**	−0.285*	0.156	0.252	−0.075	0.058	0.094	**0.352 ****	−0.253	−0.143	**−0.602 ****	1.000

** Correlation is significant at the 0.01 level (two-tailed), and * Correlation is significant at the 0.05 level (two-tailed).

**Table 4 genes-14-00155-t004:** Genetic variation analysis for 27 molecular SSRs markers studied among *Ziziphus nummularia* collected from different geographical regions of KP, Pakistan.

Locus	S. Size	BP	Obs_Het *	Exp_Het *	Nei **	PIC	GD	AF	ne *	I *	na *	FIS	FIT	FST	Nm *
**ZSSR-21**	184	3	0.88	0.6	0.6	0.59	0.47	0.85	2.51	0.99	3	−0.6	−0.5	0.03	7.2
**ZSSR-93**	268	3	0.78	0.6	0.6	0.55	0.42	0.76	2.51	0.99	3	−0.3	−0.29	0.01	33.77
**ZSSR-95**	266	5	0.95	0.76	0.76	0.75	0.61	1.1	4.13	1.49	5	−0.3	−0.26	0.05	4.75
**ZSSR-97**	222	6	0.48	0.73	0.72	0.79	0.53	0.96	3.62	1.46	6	0.17	0.33	0.19	1.06
**ZSSR-110**	226	4	0.94	0.57	0.57	0.65	0.51	0.91	2.31	0.95	4	−0.7	−0.65	0	193.4
**ZSSR-129**	158	4	0.22	0.39	0.39	0.61	0.36	0.65	1.64	0.77	4	0.41	0.46	0.09	2.56
**ZSSR-152**	250	6	0.97	0.71	0.7	0.77	0.29	0.53	3.38	1.43	6	−0.4	−0.38	0	67.41
**ZSSR-175**	250	5	0.11	0.54	0.54	0.72	0.55	0.99	2.16	0.92	5	0.82	0.82	0.01	33.35
**ZSSR-181**	198	5	0.91	0.68	0.68	0.70	0.66	1.19	3.13	1.28	5	−0.4	−0.35	0.02	11.82
**ZSSR-183**	186	4	0.87	0.57	0.56	0.67	0.66	1.19	2.29	0.94	4	−0.6	−0.6	0.01	48.42
**ZSSR-188**	298	6	0.97	0.72	0.72	0.77	0.44	0.8	3.53	1.44	6	−0.4	−0.35	0	74.54
**ZSSR-192**	238	6	0.94	0.73	0.73	0.73	0.32	0.57	3.65	1.48	6	−0.3	−0.31	0.01	38.24
**ZSSR-207**	300	4	0.95	0.62	0.62	0.54	0.72	1.3	2.64	1.05	4	−0.6	−0.53	0.01	21.39
**ZSSR-209**	190	5	0.16	0.48	0.47	0.68	0.49	0.88	1.9	0.78	5	0.68	0.69	0.05	4.49
**ZSSR-239**	236	3	0.62	0.51	0.51	0.59	0.41	0.73	2.03	0.74	3	−0.2	−0.22	0.02	14.84
**ZSSR-222**	262	5	0.33	0.59	0.59	0.72	0.58	1.04	2.44	1.18	5	0.16	0.38	0.26	0.71
**ZSSR-247**	128	3	0.44	0.53	0.52	0.59	0.65	1.17	2.1	0.9	3	0.12	0.14	0.02	13.07
**ZSSR-261**	290	3	0.34	0.31	0.3	0.47	0.51	0.92	1.44	0.58	3	−0.1	−0.13	0.01	30.84
**ZSSR-262**	278	3	0.86	0.61	0.61	0.56	0.41	0.74	2.55	1	3	−0.4	−0.4	0.01	16.51
**ZSSR-414**	200	3	0.07	0.14	0.14	0.47	0.51	0.91	1.16	0.3	3	0.47	0.5	0.05	4.51
**ZSSR-416**	138	3	0.68	0.67	0.66	0.55	0.58	1.05	2.97	1.09	3	−0.1	−0.04	0.06	3.68
**ZSSR-460**	210	4	0.92	0.69	0.69	0.65	0.34	0.61	3.22	1.26	4	−0.4	−0.34	0.01	37.52
**ZSSR-465**	248	4	0.96	0.63	0.62	0.65	0.48	0.87	2.66	1.07	4	−0.5	−0.54	0	92.14
**ZSSR-485**	176	3	0.57	0.49	0.49	0.50	0.52	0.93	1.95	0.8	3	−0.2	−0.2	0.02	16.04
**ZSSR-490**	224	2	0.21	0.24	0.24	0.33	0.29	0.52	1.31	0.4	2	0.09	0.13	0.05	4.48
**ZSSR-513**	192	4	0.88	0.73	0.73	0.65	0.28	0.51	3.68	1.34	4	−0.2	−0.2	0.03	9.48
**ZSSR-774**	300	5	0.89	0.68	0.68	0.75	0.55	0.99	3.14	1.3	5	−0.3	−0.3	0	57.96
**Mean**	**227**	**4.43**	**0.662**	**0.575**	**0.572**	**0.671**	**0.494**	**0.89**	**2.6**	**1.03**	**4.11**	**−0.2**	**−0.17**	**0.04**	**6.415**

Note: * Expected homozygosity and heterozygosity were computed using ** Nei’s (1973) expected heterozygosity * BP= Polymorphic bands * ne = Effective number of alleles, * I = Shannon’s Information index, * Nm = Gene flow estimated from FST = 0.25(1−FST)/FST.

**Table 5 genes-14-00155-t005:** AMOVA (Analysis of molecular variance) for the 180 *Z. nummularia* genotypes.

Source	Df	SS	Component of Variance	Percentage of Variation	Fixation Index	*p*-Value
Among Populations	2	82.985	0.321	8%	***FST*** 0.092	0.001
Among Individuals within Population	176	565.959	0.453	12%	***FIS*** 0.102	0.001
Within Individual	179	559.000	3.123	80%	***FIT*** 0.105	0.001
**Total**	357	1207.944	3.490	100%		

Note: df, degrees of freedom; SSD, the sum of squares; FIT, fixation index within individuals; FIS, Fixation index among populations; FST, among individuals within populations.

## Data Availability

Not applicable.

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
