# Peer review of "Characterization of the Genetic Variability within Ziziphus nummularia Genotypes by Phenotypic Traits and SSR Markers with Special Reference to Geographic Distribution"

_genes, 2023, doi:10.3390/genes14010155_

Round 1

Reviewer 1 Report

This research addressed the impacts and constraints of climate change on Z. nummularia geographical distribution is crucial for its future sustainability based on morphological and molecular data. This paper should provide valuable information for the essential roles of climatic changes on phenotypic and genetic diversity within populations. However I have some concerns :

1. 1.     the language of the manuscript is poor which influence the understanding of the presented article.  Some mistakes are indicated in the text.

1.     the overall language of the manuscript is poor which influence the understanding of the presented article.  May authors need to get help with some language editing surface.

2. No recent references were used in the manuscript, pls. try to add more recent references which are relevant to the field of this research.

3. Tables and figures are not organized well, pls. insert figures with better resolution

4. all sections of the manuscript need to be rechecked 

Reviewer 2 Report

Overall, this research article represents an interesting investigation on “Characterization of the genetic variability within Ziziphus nummularia genotypes by phenotypic traits and SSR markers with special reference to geographic distribution”. Abstract doesn’t seem logical and doesn’t provide the concise summary of the findings, The introduction doesn’t provide sufficient background of the study, the methods are generally appropriate for the experiments conducted. The analysis and results presented in figures seem logical while interpretation is supported by results. Moreover, the results are not clearly described making the manuscript hard to understand for readers. In order to improve the present study, some essential modifications have to be fixed before it proceeds, and decisive action can be taken. In addition, the study needs extensive editing on language and grammatical issues. All the comments and remarks are given below.

There are numerous flaws in the manuscript with language and writeup, manuscript needs extensive editing with language and grammar issues, some sentences are hard to understand even.

Discuss the limitations of the previous works as a motivation of the current study.

Introduction needs improvement, be specific, no need to discuss unnecessary things, introduction needs improvement with flow and consistency.

The review of the related works is weak particularly in introduction. For a broader view on the state-of-the-art, discuss the papers on the background of genetic variability within Ziziphus nummularia using different markers.

Strengthen the discussion by adding some literature about the related studies.

The figures quality needs to be improved.

There are numerous mistakes in English language and grammar. Some are mentioned here.

In line 21, “and the suitable distribution region of Swat will move to the Buner region” ????? doesn’t make sense.

In line 45, “Here, I primarily concentrate” ?????

Reviewer 3 Report

see comments in attached pdf

Round 2

Reviewer 1 Report

Authors responded properly to the addressed conerns

Reviewer 2 Report

The authors have somehow improved the manuscript; however, I can't find point to point responses to my comments and questions raised in previous revision, neither in manuscript nor in author response file. Moreover, Figures 6 & 7 are hard to read as they are out of page.